# Targeting CD4+ T Cell Glucose Metabolism: A Novel Immunotherapeutic Approach for Type 1 Diabetes

**DOI:** 10.3390/biom15060770

**Published:** 2025-05-27

**Authors:** Ying Niu, Na Wang, Leiying Qiao, Zhenxia Huang, Gaojing Jing, Songbo Fu, Xulei Tang

**Affiliations:** 1Department of Endocrinology, The First Hospital of Lanzhou University, Lanzhou 730000, China; niuy18@lzu.edu.cn (Y.N.); jinggj21@lzu.edu.cn (G.J.); 2The First Clinical Medical College, Lanzhou University, Lanzhou 730000, China; wangna2021@lzu.edu.cn (N.W.); 320220910580@lzu.edu.cn (Z.H.); 3Gansu Province Clinical Research Center for Endocrine Disease, Lanzhou 730000, China; 4The First Clinical Medical College, Nanjing University of Chinese Medicine, Nanjing 210000, China; leiyingqiao2021@njucm.edu.cn

**Keywords:** type 1 diabetes, glucose metabolism, CD4+ T cell, Th1, Th17, Tregs, targeted treatment

## Abstract

Type 1 diabetes (T1D) is a chronic autoimmune disease characterized by the destruction of insulin-producing β-cells. Emerging evidence highlights the pivotal role of CD4+ T cell subsets in driving T1D pathogenesis, where their activation, proliferation, and differentiation are tightly regulated by glucose metabolic pathways. Recent studies demonstrate that key enzymes and intermediates of glycolysis, OXPHOS, and other metabolic pathways critically modulate CD4+ T cell functions. In this review, we discuss how glucose metabolic pathways affect CD4+ T cell differentiation and functions. We also summarize the latest progress regarding glucose metabolism intervention to control the T cell immune response in T1D, with the expectation of providing new insights into T1D progression and treatment.

## 1. Introduction

Type 1 diabetes (T1D) is a chronic autoimmune disease in which the immune system attacks insulin-producing β-cells, leading to hyperglycemia [1]. Although the etiology of T1D is not completely understood, the pathogenesis of the disease is thought to involve the T-cell-mediated destruction of β-cells [2]. The latest advances in immunometabolism have shown that cellular metabolism plays a fundamental role in shaping T cell responses. T cell activation and proliferation are supported by metabolic reprogramming to meet increased energy and biomass demand, and deregulation in immune metabolism can lead to autoimmune disorders [3]. CD4+ T cells play a central role in this process. In T1D, both CD4+ and CD8+ T cells target autoantigens, leading to the inflammation and destruction of insulin-producing β-cells [4]. This destruction is mainly orchestrated by self-reactive CD4+ T cells, which are the basic drivers of disease pathology [5]. Previous studies in NOD mice have demonstrated that CD4+ T cell depletion prevents spontaneous T1D development, highlighting their pathogenic role in disease onset [6]. Therefore, elucidating the immunometabolic mechanisms of CD4+ T cells in T1D pathogenesis holds significant implications.

Immunometabolic pathways are crucial in regulating immune cell differentiation and function [7]. Emerging evidence reveals a close link between T cell responses and immunometabolism in T1D. In a resting state, T cells rely on oxidative phosphorylation (OXPHOS) and fatty acid oxidation (FAO) for energy [8]. Upon activation, they switch to aerobic glycolysis and the pentose phosphate pathway (PPP) to support growth, proliferation, and differentiation [7,9]. This metabolic shift depends on their activation status. After activation, naïve T cells will differentiate into subsets like T effector cell (Teff) and regulatory T cell (Treg) subsets [10]. These cell subtypes can mediate β-cell destruction and promote the progression of T1DM, and different subsets have unique metabolic needs that shape their differentiation and function. When the glycolytic level of CD4+ T cells increases to meet their energy and biosynthetic needs, the activation, proliferation, and survival of CD4+ T cells and the differentiation of Teffs are promoted, while the differentiation of Tregs is inhibited [11].

This review summarizes recent progress in understanding glucose metabolism in CD4+ T cells and its role in T1D. We explore how glucose metabolism regulates CD4+ T cell activation and differentiation in T1D pathogenesis and, additionally, we discuss therapeutic strategies targeting glucose metabolism in CD4+ T cell subsets, offering potential immunometabolic interventions for T1D treatment.

## 2. CD4+ T Cells in T1D: From Metabolic Dysregulation to Autoimmunity

In the resting state, the metabolic activity of naïve T cells is low, which is beneficial to their long-term survival and maintenance [8]. Remaining in this quiet phase is crucial for maintaining the diversity of T cell populations and preventing the premature activation of autoimmune responses. The maintenance of this state depends on the continuous stimulation of the T cell receptor (TCR) by the autopeptide-major histocompatibility complex (*p*-MHC) and cytokine IL-7 and requires a variety of negative regulatory mechanisms and synergistic effects of transcription factors [12]. For example, the transcription factor KLF2 maintains a resting state by inhibiting activation-related genes, while the deletion of forkhead box class O3a (Foxo3a) and Foxj1, members of the forkhead box transcription factor family, reduces the expression of NF-κB inhibitor IκB and increases the activity of NF-κB, causing T cells to overproliferate [13,14]. In addition, the maintenance of the resting state also needs to include the maintenance of the inactivated state of the activated nuclear factor of activated T cells (NFAT) and rely on the negative regulation of forkhead box protein 1 (Foxp1); Foxp1 prevents T cell activation by inhibiting the expression of proliferation-promoting genes [15,16].

When T cells recognize antigens, they will break away from the resting state to initiate clonal expansion and differentiate into effector cells [8]. This process is accompanied by a metabolic shift in glucose utilization from oxidative phosphorylation to aerobic glycolysis (Warburg effect) to meet the distinct energetic and biosynthetic demands of T cell activation, proliferation, and cytokine production. After recognizing antigens, T cell activation requires the synergy of three key signals: CD4+ T cell activation begins with the interaction between TCR and peptides loaded on MHC II, thereby initiating metabolic reprogramming on naive CD4+ T cells. The costimulatory receptors present on both APC and T cells will provide the second signal required for T cell activation. Finally, cytokines and ROS present in the local environment will provide a third proinflammatory signal, allowing T cells to differentiate into specific effector cells [17]. Mature T cells with a TCR are divided into two main subsets based on the presence of either CD4 or CD8 coreceptors [18].

T1D is characterized by the immune system specifically destroying insulin-producing β-cells. Current studies show that antigen presentation to CD4+ T cells through islet dendritic cells and macrophages is the main factor that stimulates β-cell death. CD8+ T cells drive β-cell destruction in insulitis by recognizing β-cell antigens and releasing cytotoxic molecules. B lymphocytes produce autoantibodies and regulate the activation, proliferation, and cytokine production of autoreactive CD4+ T cells. NK cells are associated with destructive infiltration, eventually leading to the progressive destruction of β-cells and insulin secretion defects [19,20]. When islet antigens are recognized, CD4 + T cells differentiate into effector subsets with different immune functions under the guidance of cytokines and transcriptional regulators, including effector T cells (Teff), follicular helper T cells (Tfh), and regulatory T cells (Treg) [21,22]. Each subset plays a unique role in the immune response: Teff cells drive β-cell death, Tfh cells support antibody production, and Tregs maintain immune tolerance. The imbalance between Tregs and Teff cells, such as T helper 1 (Th1) and Th17 cells, is widely recognized as a central mechanism in autoimmune diseases, particularly T1D [23]. This imbalance can lead to the destruction of immune homeostasis and β-cell death. The roles of different CD4+ T cell subsets in T1D progression are explored in detail below.

### 2.1. Th1 Cells: Glycolysis-Driven Inflammation and β-Cell Destruction

CD4+ T cells differentiate into Th1 and Th2 cells based on the Th1/Th2 dichotomy, which is critical for immune responses. Th1 cells promote cell-mediated immunity and Th2 cells drive humoral immunity. In 1995, Katz et al. demonstrated that T cells expressing the diabetogenic TCR trigger diabetes in neonatal NOD mice when they differentiate into a Th1 but not a Th2 phenotype, highlighting the role of Th1 cells in promoting diabetes [24]. This seminal study established the importance of Th1 cells in T1D pathogenesis and paved the way for further research into Th1-mediated autoimmunity. Studies have shown that proinflammatory cytokines such as IL-12 and IFN-γ drive CD4 + T cell differentiation toward the Th1 lineage, making the Th1 immune response dominant in the pathogenesis of T1D [22]. In T1D, Th1 cytokine responses promote inflammation and β-cell destruction through glycolysis [4,17]. Glycolysis provides the rapid energy required for Th1 cell proliferation and cytokine production, fueling the inflammatory response. Th1 cells express the transcription factor (TF) T-bet and produce interferon-gamma (IFN-γ), both of which are involved in Th1 cell responses to control cell-mediated immunity [25]. This downregulation could impair Th1 cell function, contributing to immune dysregulation. The pro-inflammatory cytokine IFN-γ plays a pivotal role in key stages of insulitis pathogenesis, mediating the destruction of pancreatic β-cells through cellular immune responses. Sun et al. demonstrated significantly elevated serum IFN-γ levels in T1D mouse models, which exacerbated inflammatory responses, thereby confirming this pathogenic mechanism [26]. Trivedi et al. previously demonstrated that the JAK1/JAK2 inhibitor AZD1480 blocked IFN-γ-mediated MHC class I upregulation on β-cells, reduced immune cell infiltration into pancreatic islets, and reversed established autoimmune insulitis in NOD mice [27]. The CD122 blockade restores immunological tolerance in autoimmune T1D via multiple mechanisms, including suppressing IFN-γ production in various pathogenic T cells and inhibiting the conversion of Th17 cells into diabetogenic Th1 cells. This finding also confirms that Th1 cells are the primary pathogenic cells of T1D, making them a promising therapeutic target [28]. Nevertheless, categorizing T1D into Th1 and Th2 modes is an oversimplification, as the developmental outcomes of specific Th cell populations undoubtedly involve collaboration among different immune cell types and factors. Their study found that, in diabetic patients receiving insulin therapy, the gene expression of T-bet and IFN-γ was significantly reduced, while the expression of GATA-3 (a key transcription factor in the early polarization of Th2 cells) was similar between patients and the control group. This suggests that changes in the gene expression of T-bet and GATA-3 mRNA may be related to the pathogenesis of T1D. Overall, this study provides evidence of Th1/Th2 imbalance in T1D patients during insulin therapy [29]. Future research is necessary to clarify the changes in T cell subpopulations associated with the duration of insulin treatment in T1D.

### 2.2. Th17 Cells: IL-17-Mediated Autoimmunity and Metabolic Reprogramming

Both human and murine Th17 cells are characterized by expression of transcription factor retinoic-acid-receptor-related orphan receptor gamma-T (ROR-γt) and secretion of IL-17A (henceforth referred to as IL-17), IL-17F, and IL-22 as hallmark cytokines [30]. IL-17A enhances the recruitment of neutrophils and other immune cells to the pancreas, exacerbating inflammation and β-cell destruction [31]. The relative role of Th17 cells in T1D pathogenesis has been suggested in both human and animal models. The investigations have used IL-17 as a surrogate marker for Th17 activity [30]. The NOD mouse model has been pivotal in demonstrating IL-17’s pathogenic role in T1D, where elevated IL-17/IL-17F expression in Langerhans islets correlates with insulitis progression and IL-17-deficient NOD mice showed delayed onset of T1D and reduced insulin inflammation [30]. In addition, a blockade with the monoclonal antibody anti-IL-17 prevents the onset of T1D in NOD mice [32]. In this study, researchers demonstrated that, mechanistically, IL-17 works synergistically with other proinflammatory cytokines to amplify the autoimmune response. These findings highlight the potential of targeting IL-17 as a therapeutic strategy for T1D. The STZ model further demonstrates the role of Th17 cells in T1D, as STZ-induced diabetes is associated with increased IL-17 production [33]. In T1D patients, IL-17 production is increased in the serum [34]. This elevation correlates with disease activity, suggesting that IL-17 could serve as a biomarker for T1D progression. There are more IL-17-positive T cells in children with new-onset and longstanding T1D than in age-matched nondiabetic controls [35]. This observation underscores the role of Th17 cells in both the initiation and perpetuation of T1D. A study of children with T1D showed that peripheral blood Th17 cell numbers were upregulated, primarily through the increased secretion of IL-17 and activation of the IL-17 pathway in vivo. This upregulation is driven by proinflammatory cytokines such as IL-6 and TGF-β, which promote Th17 differentiation. Additionally, IL-17, in synergy with IFN-γ, contributes to human β-cell death in vitro by promoting inflammatory and apoptotic responses [36]. This synergy highlights the complex interplay between Th1 and Th17 cells in T1D pathogenesis.

### 2.3. Tregs: The Metabolic Basis of Immune Tolerance and Dysfunction in T1D

Tregs are essential for maintaining peripheral immune tolerance in healthy individuals. They prevent the occurrence of autoimmune response by inhibiting the activation and proliferation of autoreactive T cells. Substantial evidence supports the notion that the uncontrolled activation and expansion of autoreactive T cells in T1D result from defects in immunosuppressive Tregs [37]. Decreased Treg cell numbers and/or function may be the primary mechanism leading to the failure of self-tolerance. For example, mutations in the Foxp3 gene, which is essential for Treg function, are associated with severe autoimmune disorders. The epigenetic modification of the Foxp3 locus can affect the stability and function of Tregs, and the hypomethylation status is related to the enhanced inhibitory activity. Foxp3 + Treg cells characterized by inhibitory function are defective or reduced in T1D [38]. This deficiency underscores the importance of Tregs in preventing autoimmune diabetes and highlights the potential of Treg-based therapies for T1D treatment. In NOD mice, impairments to Treg numbers and functionality allow autoreactive Teff cells to destroy insulin-producing β-cells, inducing spontaneous autoimmune diabetes [39,40]. Prospective clinical studies conducted by Marek-Trzonkowska et al. have shown that autologous Tregs infusion therapy is safe and well tolerated for children newly diagnosed with type 1 diabetes. This therapy can effectively inhibit the activity of autoreactive Teff, thereby delaying disease progression. However, the 1-year follow-up data showed that the therapeutic effect of Tregs would decrease over time. This phenomenon may suggest that the life cycle of in vitro expanded Tregs is short after administration, and further improvement of the treatment and validation of the inclusion criteria are needed [41,42]. Kühtreiber et al. conducted an 8-year clinical trial in which two doses of Bacillus Calmette-Guérin (BCG, an attenuated strain of Mycobacterium bovis historically used as a tuberculosis vaccine) were administered to T1D patients to trigger two clinical effects: stable and long-term reduction in blood glucose and epigenetic changes in Treg characteristic genes to restore tolerance. Both beneficial effects seem to be driven by a systemic metabolic shift from oxidative phosphorylation to accelerated and early aerobic glycolysis. The team’s subsequent research data further indicate that BCG vaccination upregulates Myc, and Myc gradually induces the glycolysis pathway in T cells and monocytes. This experiment still has some limitations; for example, the experiment still needs to be extended to more subjects of all ages and with all durations of diabetes [43,44].

## 3. Glucose Metabolism: The Key Regulation for CD4+ T Cell Differentiation in T1D

Glucose is a crucial metabolic substrate for T cells to produce energy. The glucose metabolic pathways include glycolysis and OXPHOS, which are the primary ATP-producing pathways for cells. Glucose metabolism via the OXPHOS pathway does not result in significant lactate accumulation when completing glucose oxidation. Glycolysis occurs in both aerobic and anaerobic states, providing a rapid but less efficient energy source compared to OXPHOS. During glycolysis, glucose is converted into pyruvate through a series of intermediate metabolites, which then enter the PPP and promote biosynthesis for cell growth. The pentose phosphate pathway is particularly important for generating nucleotides and maintaining redox balance, which are essential for T cell proliferation and function. In the mitochondria, pyruvate is further converted to acetyl-CoA, which enters the TCA cycle. The TCA cycle not only produces ATP but also provides intermediates for amino acid and lipid synthesis, supporting the biosynthetic demands of activated T cells. Under anaerobic conditions, glucose is metabolized to lactate through glycolysis without involving OXPHOS activity. Aerobic glycolysis (also called the Warburg effect) refers to the fact that, under conditions of sufficient oxygen, cells still preferentially convert glucose to lactic acid, rather than completely oxidizing it through the TCA cycle. This effect has been identified as a marker of activated T cells and is essential for their rapid proliferation and effector function [45,46]. In addition to energy and biosynthetic intermediates, NAD+ is reduced to NADH in glycolysis, promoting cellular redox balance [47] (Figure 1). Maintaining redox balance is critical for preventing oxidative stress and ensuring T cell survival, particularly in the inflammatory microenvironment of T1D.

### 3.1. Metabolic Features of Quiescent T Cells: Low Energy Demand and Persistence

Quiescent, nonproliferating T cells (such as naïve and memory T cells) have a unique set of metabolic adaptation mechanisms due to their nonproliferative properties and basic energy needs. Although the OXPHOS pathway has a high ATP production efficiency, its slow response rate just matches the sustained low metabolic needs of these cells. This ‘efficient and moderate’ metabolic pattern makes it an ideal choice for cells with low energy demand (such as resting T cells) [7,48]. As a master regulator of metabolism, glucose transporter 1 (GLUT1) maintains a low-level basic glucose uptake. GLUT1 expression is strictly regulated to ensure that quiescent T cells do not undergo unnecessary activation, which may lead to autoimmunity. The survival of naïve T cells depends on IL-7 signaling. IL-7 is a key cytokine that maintains T cell homeostasis by promoting survival and preventing apoptosis, ensuring a pool of naïve T cells ready to respond to antigenic stimuli. It promotes extracellular glucose uptake through GLUT1 and acts as a major regulator of metabolism through transcriptional control of the glycolytic enzyme extracellular kinase II gene. IL-7 stimulates glucose uptake by overexpressing GLUT1 and upregulating the gene encoding the glycolytic enzyme hexokinase II [48]. Hexokinase II is the first enzyme in the glycolytic pathway and plays a crucial role in glucose metabolism, ensuring that quiescent T cells have sufficient energy to respond to activation signals. This may be related to IL-7 promoting the transport of GLUT1 to the cell surface and enhancing the activity of cell surface transporters [49].

After antigen recognition, signals from TCR, coupled with appropriate co-stimulation and cytokine signals, lead to T cell proliferation and activation [48]. In contrast to resting T cells, activated, proliferating, and effector T cells have higher energy and biosynthetic requirements. Glycolysis is the primary metabolic pathway that promotes T cell activation [34]. After activation, under the condition of sufficient oxygen, T cells will still give priority to the anabolic process of glycolysis to meet the energy demand, which is a metabolic hallmark of activated T cells [50,51]. In activated T cells, GLUT1, as a major glucose transporter, drives glycolysis by enhancing glucose uptake [22]. Increased GLUT1 expression ensures that activated T cells have sufficient glucose to support their metabolic demands, particularly in the hyperglycemic environment of T1D. Specific GLUT1 inhibitors, such as STF31, WZB117, and BAY-876, are promising candidates for targeting glycolysis [9]. Glucose-induced insulin secretion from human islets was significantly impaired following STF31-induced inhibition of GLUT1 activity and STF31 also significantly inhibited glucose-stimulated ATP generation in human islets [52]. In patients with T1D, targeting glucose uptake by GLUT1 using the specific inhibitor WZB117 temporarily inhibits activated autoreactive T cells [53]. Regarding the potential side effects of GLUT1 inhibition in humans, studies on GLUT1-deficiency syndrome indicate neurological disorders in affected infants and children, while most symptoms stabilize or go into remission in adults [54]. Different CD4+ T cell subsets, such as Th1, Th17, and Tregs, have distinct metabolic requirements that shape their differentiation and function, highlighting the importance of metabolic reprogramming in immune responses. Depending on the immune response, glucose metabolic pathways through GLUT1, hypoxia-inducible factor-1α (HIF-1α), and glycolytic enzymes influence the differentiation of CD4+ T cells into Teff cells or Tregs.

### 3.2. Glycolytic Dependency of Th1 Cells: From Metabolism to Function

Glycolysis is a key pathway driving Th1 cell differentiation, upregulating T-bet and IFN-γ expression [55]. Th1 cells possess a high glycolytic rate and express high levels of GLUT1 [56]. The high glycolytic rate in Th1 cells supports their effector functions, such as cytokine production and cytotoxicity, which are critical for immune defense. Cytokine-producing Th1 cells are reduced when GLUT1 deficiency blocks glycolytic glucose transport [17]. This highlights the critical role of glucose metabolism in maintaining Th1 cell function, particularly in autoimmune diseases like T1D. In mouse T cells, transgenic overexpression of GLUT1 leads to upregulated glucose uptake and increased production of IL-2 and IFN-γ [57]. Given that Th1 cytokines (IL-2 and IFN-γ) have been shown to be associated with T1D [58], these findings suggest that enhancing glucose uptake could potentiate Th1 responses in certain contexts. However, this hypothesis requires further validation. In line with this hypothesis, increased glucose uptake by T cells via GLUT1 in patients with T1D correlates with β-cell function and disease progression. Tang et al. showed that patients with higher T cell glucose uptake had a shorter duration of diabetes. Further analysis of the follow-up data revealed that patients with low T cell glucose uptake retained an advantage in β-cell function and glycemic control trajectory. This study provides a potential marker for assessing disease prognosis and suggests that the presence of a T cell population characterized by elevated glucose metabolism enhances immune attack on β-cells in T1D [59].

HIF-1α increases glucose uptake and accelerates catabolism through glycolysis [60]. HIF-1α is stabilized under hypoxic conditions, which are common in inflamed tissues, and promotes glycolysis to support T cell function in these environments. The current research on HIF-1α regulating T cell function through glycolysis is mainly focused on Th17 cells, the regulatory mechanism of HIF-1α on Th1 cells is still unclear. It is worth noting that it has been observed that a hypoxic environment significantly inhibits the function of Th1 cells. When Th1 cells are cultured under hypoxic conditions or HIF-1α is stable, Th1 cells lose the ability to produce IFN-γ. In contrast, HIF-1α-deficient Th1 cells are insensitive to hypoxia and maintain their ability to produce IFN-γ. Furthermore, Li et al.’s study shows that high-glucose-induced HIF-1α transcriptional activation in CD4+ T cells reduces the risk of diabetic nephropathy by suppressing Th1 responses. This may provide a potential direction for future research [61].

In the activation and differentiation of T cells, it is not only glycolysis but also related enzymes that play a key role. Lactate dehydrogenase A (LDHA) converts pyruvate produced by glycolysis into lactate acid and regenerates NAD+ to sustain glycolysis. In activated T cells, induced LDHA contributes to aerobic glycolysis and supports IFN-γ expression [17,51]. LDHA activity is critical for Th1 cell effector functions, and its inhibition could be a therapeutic strategy to modulate Th1-mediated autoimmunity. In mouse T cells, the ablation of LDHA protects mice from immunopathology induced by IFN-γ overexpression [51]. The glyceraldehyde-3-phosphate dehydrogenase (GAPDH) plays a key role in glycolysis and gluconeogenesis. During T cell activation, GAPDH is actively engaged in glycolysis, and GAPDH acetylation positively regulates GAPDH activity and Th1 differentiation [60]. Activated protein kinase (AMPK) orchestrates metabolic reprogramming to stimulate Teff cell responses during nutrient starvation and mitochondrial homeostasis [62]. In addition, 5′AMPKα1 deficiency impairs the expansion of Th1 cells in lymphopenic environments [63]. This highlights the importance of metabolic enzymes in shaping T cell fate and function. PFK15 is a competitive inhibitor of the rate-limiting glycolysis enzyme 6-phosphofructo-2-kinase/fructose-2,6- biphosphatase 3 (PFKFB3). The results of Martins et al. confirmed that PFK15 inhibits glycolysis during CD4 + T cell activation, reduces T cell response to β-cell antigens in vitro, and delays T1D seizures in vivo. These important findings reveal that precise regulation of the PFKFB3-mediated glycolysis pathway can encourage autoreactive T cells to enter a state of end-depletion, thus providing a new targeted intervention strategy for T1D immunotherapy [64].

The mechanistic target of rapamycin (mTOR) signaling pathway is a key regulator of the development of specific types of Teff cells [34]. mTOR complex 2 (mTORC2) activity induces glycolysis by promoting Akt activity, which upregulates the surface expression of GLUT1, thereby regulating Th1 cell differentiation [62]. mTORC2-mediated Akt activation enhances glucose uptake and supports Th1 cell effector functions, highlighting the potential of targeting mTOR signaling to modulate immune responses in T1D. Therefore, reducing glycolysis and decreasing autoreactive Th1 cells provides a potential therapeutic direction to delay the onset of T1D.

### 3.3. Metabolic Regulation of Th17 Cells: Synergy Between HIF-1α and RoR-γt

Th17 cells, similar to Th1 cells, depend on glycolysis for their differentiation. Glycolysis provides the energy and biosynthetic intermediates necessary for Th17 cell proliferation and cytokine production, which are critical for their role in autoimmune diseases. Th17 cells also promote GLUT1 expression, increasing the glycolytic rate [55]. GLUT1 upregulation ensures that Th17 cells have sufficient glucose to support their metabolic demands, particularly in the inflammatory microenvironment of T1D. HIF-1α is a key regulator of cellular adaptation to hypoxia and promotes glycolysis to support Th17 cell function. HIF-1α increases the glycolysis rate in Th17 cells and induces the expression of the Th17 master transcription factor ROR-γt, which directs the differentiation program of Th17 cells [55,65,66]. ROR-γt is essential to produce IL-17, a key cytokine in Th17-mediated immunity, which contributes to β-cell destruction in T1D.

After TCR activation and the co-stimulation of naïve T cells, mTOR is stimulated by PI3K/Akt signaling, which activates HIF-1α to support glycolysis and drive Th17 differentiation [67]. Fluvoxamine is a selective serotonin reuptake inhibitor (SSRI) which significantly increases the level of serotonin in the synaptic cleft by blocking the serotonin reuptake of presynaptic nerve endings. In the NOD mouse model, Fluvoxamine treatment significantly delayed disease progression, manifested as reduced insulin resistance and improved islet β-cell function. Mechanism studies have shown that Fluvoxamine downregulates the glycolysis process by inhibiting the PI3K–AKT signaling pathway, thereby inhibiting Th1 and Th17 polarization and function. This study provides an important theoretical basis for the repositioning of SSRIs in autoimmune diseases [68]. mTOR signaling integrates metabolic and immune signals to promote Th17 cell development, highlighting its potential as a therapeutic target in autoimmune diseases [69]. The liver kinase B1 (LKB1) is a key metabolic regulatory kinase that inhibits mTOR signaling by activating the AMPK pathway and plays a central role in maintaining cell energy homeostasis, regulating cell growth and metabolic reprogramming. PTEN is an upstream regulator of mTORC1 independent of AMPK. Mahesh et al. found that, in the absence of inflammation, LKB1-deficient T cells preferentially differentiate into Th1 and Th17 cells. Further studies have confirmed that LKB1 controls mTORC1 signaling through PTEN activation rather than AMPK and acts as a key upstream regulator of Th1/17 differentiation by inhibiting the mTORC1–HIF-1α–glycolysis cascade [70]. This research highlights the importance of maintaining immune homeostasis and its potential as a therapeutic target in T1D. Manganese metalloporphyrin (MnP) is an effective antioxidant. Studies have shown that reactive oxygen species are required to drive efficient and sustained aerobic glycolysis during CD4+ T cell activation, and MnP treatment can significantly inhibit this mTOR-driven aerobic glycolysis in CD4+ T cells. In the adoptive transfer model of T1D, animals treated with MnP showed delayed progression of diabetes, accompanied by the decreased activation of CD4+ T cells. This study not only reveals the key role of ROS–mTOR–glycolysis axis in autoimmune diseases but also provides a new metabolic intervention target for the treatment of autoimmune diseases such as T1D [71].

### 3.4. Metabolic Uniqueness of Tregs: Fatty Acid Oxidation and Immunosuppressive Function

The metabolic pathways in Treg development differ from those in Teff cells. Tregs rely on oxidative metabolism to support their long-term survival and suppressive function, which is critical for maintaining immune tolerance [34]. Fatty acid oxidation (FAO) is highly efficient in generating ATP, making it ideal for Tregs with low proliferative rates. However, glycolysis is essential for the differentiation of human Foxp3+ Tregs, as the induction and suppressive function of induced Tregs (iTregs) closely rely on glycolysis. Glycolysis provides the necessary energy and intermediates for Treg induction and function, highlighting the metabolic flexibility of Tregs. Glycolysis inhibits Foxp3 splicing variants containing exon 2 (Foxp3-E2) via the glycolytic enzyme enolase-1, and this finding was confirmed in T1D patients [72]. This mechanism ensures that Tregs maintain their suppressive phenotype, which is critical for preventing autoimmunity.

The high expression of activated AMPK in Tregs can control mTORC1, reduce GLUT1 expression, promote FAO, and inhibit glycolysis, which is conducive to the differentiation and function of Treg cells and the maintenance of immune tolerance. Therefore, Tregs exhibit lower expression levels of GLUT1 and higher rates of lipid oxidation [73]. This metabolic profile distinguishes Tregs from proinflammatory Teff cells and supports their immunosuppressive function, particularly in the context of T1D. MOTS-c is a mitochondrial-derived peptide (MDP) that promotes metabolic homeostasis in an AMPK-dependent manner. Studies in NOD mice have found that MOTS-c can significantly improve the progression of hyperglycemia and reduce the number of infiltrating immune cells in islets. Furthermore, MOTS-c reduced T cell activation by alleviating T cells from the glycolytic stress in T1D patients suggesting a potential therapeutic implication. Metabolic and genomic analysis revealed that MOTS-c can inhibit TCR/mTORC1 signaling and regulate T cell differentiation to support Treg cells. These findings suggest that MOTS-c has the potential to prevent autoimmune diabetes [74].

LKB1 acts as a metabolic sensor that regulates Treg stability and function, ensuring that Tregs can effectively suppress immune responses [75]. LKB1 deficiency impairs Treg differentiation and leads to autoimmunity, highlighting its importance in immune regulation. GLUT1 expression can increase the number of Tregs but decreases their suppressive effect and Foxp3 expression [76]. This suggests that excessive glycolysis may compromise Treg function, which could contribute to the loss of immune tolerance in T1D. HIF-1α is a key metabolic sensor that regulates the balance between Treg and Th17 differentiation. Deng et al. found that HIF-1α regulates Th17 signature genes to enhance TH17 development, while, at the same time, attenuating Treg development by binding to Foxp3 and targeting it for proteasomal degradation [77]. And Shi et al. demonstrated that HIF-1α-dependent metabolic reprogramming contributes to this cell fate decision. They found that blocking glycolysis, in turn, reduced TH17 differentiation but promoted Treg cell differentiation. Thus, HIF1α-dependent transcriptional program was important for mediating glycolytic activity, thereby contributing to the lineage choices between TH17 and Treg cells [78]. mTORC1 promotes glucose metabolism via the modulation of HIF-1α and myelocytomatosis oncogene (Myc), preventing Foxp3 expression and inhibiting Treg differentiation and maintenance [47,79]. Targeting mTORC1 could enhance Treg-mediated immune tolerance in autoimmune diseases like T1D, offering a potential therapeutic strategy to restore immune balance. The glycolysis inhibitor 2-deoxy-D-glucose (2-DG) is a glucose analog and an inhibitor of hexokinase and phosphoglucoisomerase. 2-DG has been used to study the suppressive function of glycolysis in T cells and selectively inhibits Teff cells with upregulated glycolytic activity [80,81,82]. When prediabetic NOD mice were treated with 2-DG to control aerobic glycolysis, the number of activated T cells infiltrating the islets was reduced, and β-cell granularity was improved [83]. The combination of 2-DG and metformin inhibits the CD4+ Teff cell response while inducing Tregs [48]. Strikingly, when co-administered, these treatments synergistically inhibit the upregulation of Myc, a pivotal regulator of glycolytic metabolic programs [82]. Rapamycin, an mTOR inhibitor, enhances the proliferation and regulates the function of Tregs while decreasing the proliferation of Th1 and Th17 cells in humans and mice with T1D [3,84,85]. Rapamycin directly influences human Treg function in vivo and promotes the expansion of CD4+ CD25+ Foxp3+ Tregs in mice and human models, recalibrating their suppressive activity, which is critical for maintaining peripheral tolerance homeostasis [3]. A phase 2, single-center, randomized, double-blind, placebo-controlled study in patients with longstanding T1D demonstrated that rapamycin restored Treg function and reduced insulin requirements in vivo [84] (Table 1, Figure 2).

## 4. Conclusions and Future Perspectives

The relationship between glucose metabolic pathways and immune cells, especially T cells, has been of great interest. With further research, metabolic reprogramming has been recognized as a key driver of T cell fate and function. The association between T cells and glucose metabolism in T1D provides the possibility of using T cell metabolism as a biomarker and potential therapeutic target. For instance, the upregulation of glycolysis in autoreactive T cells could serve as a diagnostic marker for early T1D progression, while metabolic inhibitors could be used to modulate pathogenic T cell responses. Activated immune cells undergo a shift in glucose metabolism, and targeting glucose metabolic pathways to manipulate T cells in T1D is a promising strategy. This approach could potentially restore immune balance by suppressing autoreactive T cells while promoting Treg function, thereby preserving β-cell mass and function. Identifying relevant target indicators related to glucose metabolism could provide new therapeutic opportunities for T1D. For example, metabolic enzymes such as hexokinase, LDHA, and glucose transporters like GLUT1 have emerged as key regulators of T cell metabolism and could be targeted to modulate immune responses.

However, our current understanding of the interplay between glucose metabolism and T cell biology in T1D remains incomplete. While significant progress has been made in elucidating the metabolic requirements of different T cell subsets, such as Th1, Th17, and Tregs, the metabolic adaptations of other immune cells, such as B cells and macrophages, in the context of T1D are less well understood. Moreover, the metabolic microenvironment in T1D, characterized by hyperglycemia and insulin deficiency, may further complicate the metabolic regulation of immune cells. For instance, hyperglycemia can exacerbate glycolytic activity in autoreactive T cells, while insulin deficiency may impair the metabolic flexibility of Tregs, leading to a loss of immune tolerance. Therefore, future studies should aim to unravel the metabolic interactions between immune cells and their microenvironment in T1D, as well as the impact of metabolic interventions on systemic immune homeostasis.

Therapeutic strategies targeting glucose metabolism in T cells have shown promising results in preclinical models of T1D. However, the translation of these findings into clinical practice requires further optimization to ensure efficacy and minimize off-target effects. For instance, the potential neurological side effects of GLUT1 inhibition highlight the need for targeted delivery systems or combination therapies to enhance specificity. Furthermore, the heterogeneity of T cell responses in T1D patients underscores the importance of personalized metabolic interventions tailored to individual metabolic and immune profiles.

In conclusion, targeting glucose metabolism in T cells represents a promising avenue for T1D therapy, with the potential to modulate immune responses and preserve β-cell function. Future research should focus on identifying novel metabolic targets, optimizing existing therapeutic strategies, and exploring the interplay between metabolism and immunity in T1D. By integrating metabolic and immunological approaches, we can develop more effective and precise therapies to halt disease progression and improve outcomes for patients with T1D.

## Figures and Tables

**Figure 1 biomolecules-15-00770-f001:**
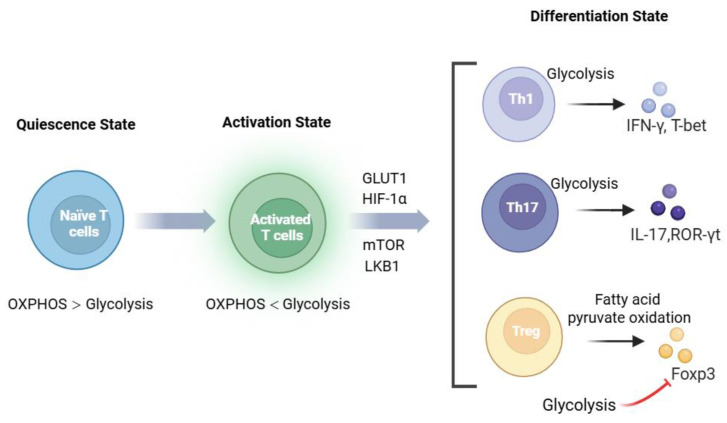
Metabolic reprogramming and pathways governing CD4+ T cell differentiation. In the quiescent state, resting T cells primarily rely on oxidative phosphorylation (OXPHOS) to generate ATP. Upon antigen recognition, CD4+ T cells exit quiescence and reprogram their glucose metabolism, shifting from OXPHOS to aerobic glycolysis to meet energy demands, mediated by regulators, including glucose transporter 1 (GLUT1), hypoxia-inducible factor 1-alpha (HIF-1α), mechanistic target of rapamycin (mTOR), liver kinase B1 (LBK1), and glycolytic enzymes. CD4+ T cell subsets exhibit distinct metabolic dependencies: Th1 cells utilize glycolysis to upregulate T-bet and IFN-γ; Th17 cells depend on glycolysis and require ROR-γt and IL-17 for differentiation and function; and Tregs maintain suppression via fatty acid oxidation (FAO) and pyruvate oxidation, with glycolysis indirectly stabilizing their phenotype by suppressing the Foxp3. This metabolic flexibility ensures functional specialization of CD4+ T cell subsets in immune responses. Created in Biorender. Ying Niu. (2025) https://app.biorender.com/illustrations/64abd31edf27d21e5ba99dc2.

**Figure 2 biomolecules-15-00770-f002:**
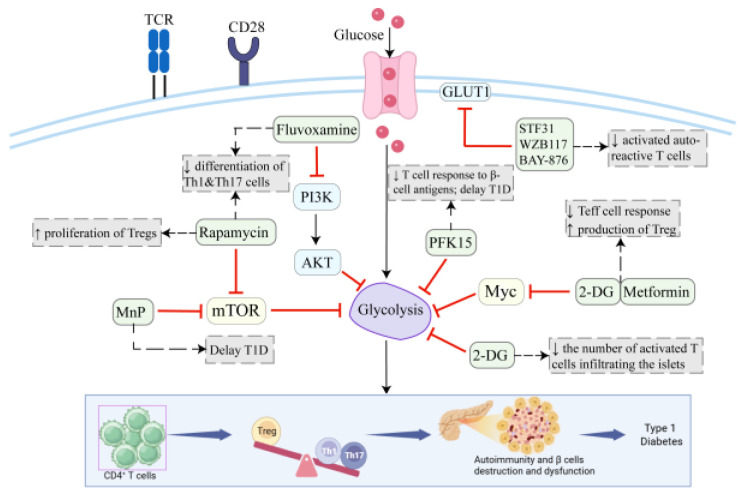
Targeting metabolic pathways: novel therapeutic strategies for T1D. In T1D, effector T cells (Teff, e.g., Th1 and Th17 cells) drive β-cell destruction, while regulatory T cells (Tregs) maintain immune tolerance. The imbalance between Tregs and Teff cells (especially Th1/Th17) lead to the destruction of immune homeostasis and the destruction and dysfunction of β-cells, eventually leading to T1D. Therapeutic strategies targeting metabolic reprogramming include (1) PFK15 inhibits glycolysis during CD4 + T cell activation to reduce T cell response to β-cell antigens and delay T1D onset; (2) glycolysis inhibition via 2-deoxy-D-glucose (2-DG), which selectively suppresses Teff activity, reduces islet-infiltrating T cells, and synergizes with metformin to inhibit the glycolytic regulator Myc; (3) GLUT1 blockade (e.g., STF31, WZB117, and BAY-876) to transiently inhibit autoreactive T cells by limiting glucose uptake through GLUT1; (4) rapamycin inhibits the mTOR pathway to enhance the proliferation of Tregs and regulate their function to maintain peripheral tolerance homeostasis.; (5) Fluvoxamine inhibits Th1 and Th17 polarization and functions by inhibiting glycolysis, thereby alleviating autoimmune progression in T1D; and (6) MnP inhibits mTOR-driven aerobic glycolysis in CD4+ T cells and delays the progression of diabetes. Created in Biorender. Ying Niu. (2025) https://app.biorender.com/illustrations/64af5668df6b1a0e8640386a.

**Table 1 biomolecules-15-00770-t001:** Targeting CD4^+^ T cell glucose metabolic pathways in T1D.

Targeting Treatment	Target	Mechanisms	Effect	Drug Development	Reference
PFK15	Glycolysis	Glycolysis inhibition prevents full T cell activation	Reduces T cell response to β-cell antigens and delays T1D seizures	Preclinical	[64]
2-DG	Glycolysis	Inhibits phosphorylation of glucose	Decreases autoreactive antigen-specific T cells and infiltration of islets	Preclinical	[83]
2-DG + Metformin	Myc	Inhibitor of mTOR	Reduces Th1 and Th17 cell response, increases Tregs proliferation	Preclinical	[48,82]
STF31,BAY-876, WZB117	GLUT1	Specific GLUT1 inhibitor	Inhibits insulin secretion; inhibits glucose-stimulated ATP production; temporarily inhibits activated autoreactive T cells	Clinical/ preclinical	[9,52,53]
Rapamycin	HIF-1α	Inhibitor of mTOR	Enhances the proliferation of Tregs, reduces the proliferation of Th1 and Th17 cells	Approved for clinical use	[3,84]
Fluvoxamine	PI3K-AKT	Downregulated glycolysis	Decreases insulin resistance and improve islet beta cell function	Preclinical	[68]
MnP	mTOR	Inhibit glycolysis	Delays T1D progression and reduces CD4+ T cell activation	Preclinical	[71]

## Data Availability

No new data were created or analyzed in this study.

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
