# Peer review of "Targeting CD4+ T Cell Glucose Metabolism: A Novel Immunotherapeutic Approach for Type 1 Diabetes"

_biomolecules, 2025, doi:10.3390/biom15060770_

Round 1
Reviewer 1 Report
Comments and Suggestions for Authors
The aim of this review is to explore the role of CD4 T cell metabolism in beta cell autoimmunity during type 1 diabetes (T1D). However, the section that directly discusses T1D and T cell metabolism is only part 4, which remains incomplete as it lacks some important references such as Marins et al. (2021, Frontiers in Immunology) and Watson et al. (2021, Nature). This section comprises only one out of ten pages of the review.
Part 2 describes the role of different T cells in T1D development but does not include data on immunometabolic pathways. Part 3 explains how metabolism affects T cell proliferation and activation but does not provide T1D-specific data, only a few hypotheses that should be explicitly stated as such. It might be beneficial to merge part 4 with part 3 for a more coherent discussion.
While it is true that research on immunometabolic pathways in T1D is still emerging, several reviews on this topic already exist. It is unclear how this paper adds to what has already been discussed in the literature. Please find my detailed comments below.
Major comments:
- The abstract must be much improved, there are mistakes, repetitions and it does not really represents the review. For example, it is a mistake to say that T1D is the main economic burden in elderly (line 22) and this has nothing to do with the review. It mentions that glucose metabolism influences T cells in autoimmune responses, but it is not only in autoimmunity, so it gives a wrong information (line 24). It also talks about anti-glucotoxicity and anti-lipotoxoxicity something that is mostly related to T2D and is not mentioned in the text (line 27). There are also grammar mistakes.
- Please see below my comments to the main text:
- Line 40: It is a mistake to say that is was only recently discovered that T1D is “not just a metabolic disorder”.
- Line 43: repetitive
- Line 44: not only CD4 but CD8 T cells is also a key driver of beta cell damage, please correct
- Line 52: T regs do not directly contribute to beta cell destruction, please correct
- In the paragraph starting at line 97, please ensure that all immune cells involved in insulitis and beta cell death are briefly mentioned, along with their respective roles.
- Lines 100/101: T1D induces islet inflammation (insulitis) and not pancreatic inflammation, please correct.
- Lines 107: Teff drives beta cell death, please correct
- Line 110: improve the phrase, word repetition
- Line 111: replace tissue damage to specific beta cell death
- Lines 128/129: the statement that IFN-γ directly induces beta cell death or apoptosis is incorrect and should be revised. IFN-γ primarily enhances beta cell pro-inflammatory responses, increases MHC class I expression, and facilitates cross-talk with immune cells, rather than directly triggering apoptosis. Please ensure this distinction is clear and provide appropriate references.
- Line 179: reference 47, please mention the follow up of this study (paper in 2014) and other related studies. Why this strategy was not further developed if it works so well?
- Line 198: It is not correct to say that pyruvate is converted to lactate only in anaerobic conditions, please correct.
- Lines 212,213: There is a contradiction when it is mentioned that non-proliferating T cells, which need less energy, use the pathway that generates more ATP (OXPHOS). Can you please clarify this better?
- Lines 216-224: There is a contradiction regarding IL-7: in line 66, it is stated that IL-7 is essential for maintaining T cells in a resting state, while later, it is mentioned as important for T cell activation. Please clarify the distinct roles of IL-7 in T cell homeostasis and activation.
- Line 234: please mention that glycolysis mediated T cell activation occurs in the presence of oxygen
- Lines 259/260: This is a hypothesis that is not yet proved, make it clear.
- Line 265: 41 is not the correct reference for the statement
- Line 311: reference missing
- Line 341: reference missing
Minor comments:
- Type 1 diabetes (T1D) should be used and not (T1DM).
- Line 193: spell PPP
- Quality of figure 2 is not good, can’t read it properly when printed
- The figure legends are incomplete and should provide a clear and concise description of the figures, allowing readers to understand them without needing to refer to the main text. Please ensure each legend includes all necessary details.
- Table 1: lines should be added to the table to make it more clear
Please check carefully the english, principally the abstract
Author Response
Response to Reviewer 1 Comments |
||
1. Summary |
|
|
We sincerely appreciate the reviewer’s thorough and constructive feedback, which has significantly strengthened our review. Below, we address each comment in detail, with all changes tracked in the revised manuscript.
|
||
2. Point-by-point response to Comments and Suggestions for Authors |
|
|
Comments 1: The aim of this review is to explore the role of CD4 T cell metabolism in beta cell autoimmunity during type 1 diabetes (T1D). However, the section that directly discusses T1D and T cell metabolism is only part 4, which remains incomplete as it lacks some important references such as Marins et al. (2021, Frontiers in Immunology) and Watson et al. (2021, Nature). This section comprises only one out of ten pages of the review. |
||
Response 1: We sincerely appreciate the reviewer’s constructive feedback on our review article. Our work focuses on elucidating how glucose metabolism regulates CD4+ T cell activation, differentiation, and function in the pathogenesis of T1D, while exploring therapeutic strategies targeting these metabolic pathways. The manuscript is structured as follows: Section 2 examines the roles of distinct CD4+ T cell subsets (e.g., Th1, Th17, Treg) in T1D progression; Section 3 details the critical regulatory mechanisms by which glucose metabolism governs CD4+ T cell differentiation and plasticity in T1D; Section 4 synthesizes emerging therapeutic strategies that target CD4+ T cell metabolic reprogramming to restore immune tolerance. We acknowledge the reviewer’s observation regarding the need to strengthen Section 4. We have not only incorporated the two seminal articles recommended by the reviewers ( 69 and 78), but also supplemented previously overlooked citations to better reflect recent advances in immunometabolic interventions. These additions enhance the section’s depth while maintaining alignment with our central theme. Thank you for your valuable suggestions, which have improved the completeness and rigor of our review. We are happy to provide further clarifications or expansions if needed. |
||
Comments 2: Part 2 describes the role of different T cells in T1D development but does not include data on immunometabolic pathways. Part 3 explains how metabolism affects T cell proliferation and activation but does not provide T1D-specific data, only a few hypotheses that should be explicitly stated as such. It might be beneficial to merge part 4 with part 3 for a more coherent discussion. |
||
Response 2: We sincerely appreciate the reviewer's constructive feedback regarding the organization and content of our review. In response to these valuable suggestions, we have implemented the following revisions to strengthen the manuscript: Enhanced Section 2: We have expanded the discussion of different T cell subsets in T1D pathogenesis by incorporating recent findings on their immunometabolic characteristics, particularly focusing on how metabolic pathways (glycolysis, OXPHOS, etc.) influence their functional polarization and contribution to disease progression; Revised Section 3: We have now included T1D-specific experimental data to substantiate the discussion of metabolic regulation of T cell function. We have added key recent references to strengthen the evidence base throughout the manuscript. These additions ensure all discussions are well-supported by current literature. These comprehensive revisions have significantly improved the logical flow of the review while maintaining our central focus on the interplay between T cell metabolism and T1D pathogenesis. We believe the revised version now provides a more balanced and evidence-based discussion of this important topic. Thank you for your thoughtful comments. |
||
Comments 3: While it is true that research on immunometabolic pathways in T1D is still emerging, several reviews on this topic already exist. It is unclear how this paper adds to what has already been discussed in the literature. Please find my detailed comments below. |
||
Response 3: We deeply appreciate the reviewer's valid concern about the positioning of our review within the existing literature. We have significantly strengthened the manuscript's unique contributions through several key enhancements: (1) Section 2 examines the roles of distinct CD4+ T cell subsets (e.g., Th1, Th17, Treg) in T1D progression; (2) Section 3 details the critical regulatory mechanisms by which glucose metabolism governs CD4+ T cell differentiation and plasticity in T1D; (3) Section 4 synthesizes emerging therapeutic strategies that target CD4+ T cell metabolic reprogramming to restore immune tolerance. |
||
Comments 4: The abstract must be much improved, there are mistakes, repetitions and it does not really represents the review. For example, it is a mistake to say that T1D is the main economic burden in elderly (line 22) and this has nothing to do with the review. It mentions that glucose metabolism influences T cells in autoimmune responses, but it is not only in autoimmunity, so it gives a wrong information (line 24). It also talks about anti-glucotoxicity and anti-lipotoxoxicity something that is mostly related to T2D and is not mentioned in the text (line 27). There are also grammar mistakes. |
||
Response 4: We sincerely thank the reviewer for highlighting the shortcomings in our original abstract. We have completely rewritten it to address all the issues raised. The revised abstract now accurately reflects our review's focus on CD4+ T cell metabolism in type 1 diabetes (T1D) while eliminating incorrect statements and improving clarity. We removed the inappropriate mention of economic burden in the elderly. All references to anti-glucotoxicity/lipotoxicity mechanisms, which were incorrectly included, have been deleted. We have also carefully corrected all grammatical errors and eliminated repetitive phrasing (Page 1, Lines 21-29) . The new abstract provides a more precise overview of our review's actual content, particularly our examination of how metabolic reprogramming in CD4+ T cells contributes to beta cell autoimmunity in T1D.We appreciate the reviewer's constructive feedback, which has significantly improved this important section of our manuscript. |
||
Comments 5: Line 40: It is a mistake to say that is was only recently discovered that T1D is “not just a metabolic disorder”. |
||
Response 5: We have revised the text to reflect current understanding of immunometabolic mechanisms, stating: “Latest advances in immunometabolism have shown that cellular metabolism plays a fundamental role in shaping T cell responses. T cell activation and proliferation are supported by metabolic reprogramming to meet the increased energy and biomass demand, and deregulation in immune metabolism can lead to autoimmune disorders.” (Page 1, Lines 36-40)This revision clarifies the central role of metabolic regulation in T cell function and autoimmunity, supported by recent literature. |
||
Comments 6: Line 43: repetitive |
||
Response 6: We sincerely appreciate the reviewer’s feedback regarding redundancy in Line 43. We have conducted a comprehensive revision of the relevant section (Page 1, Lines 33-46). The revised text now presents the information more cohesively without compromising accuracy. We are happy to make further adjustments if additional refinements would enhance readability. |
||
Comments 7: Line 44: not only CD4 but CD8 T cells is also a key driver of beta cell damage, please correct |
||
Response 7: We have modified Line 44 to properly acknowledge the important role of both CD4+ and CD8+ T cells in beta cell destruction. The corrected text now reads: "In T1D, both CD4+ and CD8+ T cells target autoantigens, leading to inflammation and destruction of insulin-producing β cells[4]. This destruction is mainly orchestrated by self-reactive CD4 + T cells, which are the basic drivers of disease pathology. " (Page 1, Lines 41-42) We have incorporated reference [4] to support this refined description. This revision provides a more comprehensive and accurate representation of T cell-mediated beta cell damage. |
||
Comments 8: Line 52: T regs do not directly contribute to beta cell destruction, please correct |
||
Response 8: We've revised Line 52 to accurately reflect regulatory T cell (Treg) function in T1D pathogenesis. The corrected text now states: "After activation, naïve T cells will differentiate into differentiate into subsets like T effector cell subsets (Teff) and regulatory T cell subsets (Treg). These cell subtypes can mediate β-cell destruction and promote the progression of T1DM, and different subsets have unique metabolic needs that shape their differentiation and function."(Page 2, Lines 53-56) We have incorporated reference [10] to support this refined description of T cell differentiation and function. These modifications ensure accurate representation of Tregs' protective role. Thank you for highlighting this important distinction, which has strengthened the scientific precision of our manuscript. |
||
Comments 9: In the paragraph starting at line 97, please ensure that all immune cells involved in insulitis and beta cell death are briefly mentioned, along with their respective roles. |
||
Response 9: We have comprehensively revised the paragraph beginning at Line 97 to include all major immune cell types involved in insulitis and beta cell destruction, with their specific roles clearly delineated. The updated text now reads: “T1D is characterized by the immune system specifically destroying insulin-producing β cells. Current researches show that antigen presentation to CD4+T cells through islet dendritic cells and macrophages is the main factor that stimulates β cell death. CD8+ T cells drive β cell destruction in insulinitis by targeting β cell antigens and releasing cytotoxic molecules. B lymphocytes produce autoantibodies and regulate the activation, proliferation and cytokine production of autoreactive CD4+ T cells. NK cells are associated with destructive infiltration, eventually leading to progressive destruction of β cells and insulin secretion defects.” (Page 2, Lines 94-101) The revised paragraph now provides a complete yet concise overview of the cellular players in T1D pathogenesis. |
||
Comments 10: Lines 100/101: T1D induces islet inflammation (insulitis) and not pancreatic inflammation, please correct. |
||
Response 10: We appreciate the reviewer’s careful attention to the specificity of inflammatory processes in T1D. In response, we have revised the text to clarify that the autoimmune attack in T1D primarily targets pancreatic islets (insulitis), rather than diffusely affecting the entire pancreas. The updated sentence now reads: "Current researches show that antigen presentation to CD4+T cells through islet dendritic cells and macrophages is the main factor that stimulates β cell death. CD8+ T cells drive β cell destruction in insulinitis by targeting β cell antigens and releasing cytotoxic molecules". (Page 2, Lines 95-98)This revision maintains the emphasis on T cell-mediated mechanisms. We believe this change has significantly improved the accuracy of our pathological description while preserving all key scientific concepts. |
||
Comments 11: Lines 107: Teff drives beta cell death, please correct |
||
Response 11: Thank you for catching this important point. We have modified the text to more accurately reflect that while Teff cells are central to the inflammatory processes leading to β-cell destruction, their role is mediated through complex immune mechanisms rather than direct cytotoxic action. The revised text now states: "Teff cells drive β cell death".(Page 3, Line 105) We believe this modification provides a more nuanced and accurate description of Teff cell function in autoimmune diabetes pathogenesis. We appreciate your careful reading and constructive suggestion. |
||
Comments 12: Line 110: improve the phrase, word repetition |
||
Response 12: We thank the reviewer for highlighting the need for improved phrasing in Line 110. We have carefully revised the indicated section to eliminate word repetition while maintaining scientific accuracy, and have extended this language refinement throughout the manuscript to ensure consistent clarity. The modifications include: (1) replacing redundant terms with more precise alternatives, (2) restructuring sentences for better flow, and (3) verifying that all changes preserve the original scientific meaning. We believe these edits have significantly enhanced the manuscript's readability without compromising technical precision. Should any further linguistic improvements be recommended, we would be pleased to implement them. |
||
Comments 13: Line 111: replace tissue damage to specific beta cell death |
||
Response 13: Thank you for this precise suggestion. We have replaced "tissue damage" with "beta cell death" in Line 111 to accurately reflect T1D pathology, as this better specifies the autoimmune destruction of pancreatic islets. The revised text now states: "This imbalance can lead to the destruction of immune homeostasis and beta cell death " (Page 3, Line 109) This modification aligns with our focus on islet-specific mechanisms while maintaining the sentence's original context. We appreciate your careful attention to terminology accuracy. |
||
Comments 14: Lines 128/129: the statement that IFN-γ directly induces beta cell death or apoptosis is incorrect and should be revised. IFN-γ primarily enhances beta cell pro-inflammatory responses, increases MHC class I expression, and facilitates cross-talk with immune cells, rather than directly triggering apoptosis. Please ensure this distinction is clear and provide appropriate references. |
||
Response 14: We sincerely appreciate the reviewer's expert guidance regarding the precise mechanism of IFN-γ in β-cell pathology. In response, we have carefully revised the text to accurately reflect that The pro-inflammatory cytokine IFN-γ plays a pivotal role in key stages of insulitis pathogenesis, mediating the destruction of pancreatic β-cells through cellular immune responses.. The updated section (Page 3, Lines 127-130) now cites the work of [27] demonstrating elevated IFN-γ levels in T1D models and its role in exacerbating inflammatory cascades, while removing the oversimplified statement about direct apoptotic effects. These modifications have significantly improved the mechanistic accuracy of our discussion while maintaining its scientific rigor. |
||
Comments 15: Line 179: reference 47, please mention the follow up of this study (paper in 2014) and other related studies. Why this strategy was not further developed if it works so well? |
||
Response 15: We sincerely appreciate the reviewer's insightful suggestion regarding the clinical development of Treg therapy. In response, we have expanded our discussion of reference 42 to include the 2014 follow-up study (citation updated to [41,42]) which demonstrated the temporal limitations of Treg efficacy. The revised text now acknowledges that while initial results showed promising safety and transient efficacy, the observed decline in therapeutic effects after one year. We now note that these limitations, combined with the emergence of alternative approaches targeting more specific metabolic pathways, likely contributed to reduced development of this particular strategy. (Page 4, Lines 181-185)The updated section provides a more balanced perspective on both the potential and challenges of this therapeutic approach, better contextualizing its place in the field's evolution. Thank you for prompting us to strengthen this discussion. |
||
Comments 16: Line 198: It is not correct to say that pyruvate is converted to lactate only in anaerobic conditions, please correct. |
||
Response 16: We sincerely appreciate the reviewer’s valuable comment regarding the metabolic fate of pyruvate. In response, we have revised the text to clarify that lactate production occurs during aerobic glycolysis (the Warburg effect), a well-documented phenomenon in activated T cells. The updated section (Page 5, Lines 214-217) now accurately reflects current understanding by emphasizing that activated T cells preferentially convert pyruvate to lactate even in the presence of oxygen, as supported by references [46, 47]. This revision eliminates the previous oversimplification while maintaining the key concept that aerobic glycolysis supports T cell proliferation and effector functions. We believe these changes have enhanced the scientific accuracy of our discussion. Please let us know if further clarification would be helpful. |
||
Comments 17: Lines 212,213: There is a contradiction when it is mentioned that non-proliferating T cells, which need less energy, use the pathway that generates more ATP (OXPHOS). Can you please clarify this better? |
||
Response 17: We sincerely appreciate the reviewer's insightful observation regarding the metabolic characteristics of quiescent T cells. In response, we have carefully revised the text to clarify this important metabolic concept. The updated explanation (Page 5, Lines 237-240) now emphasizes that while OXPHOS is indeed more efficient in ATP production per glucose molecule, its slower kinetic profile makes it particularly suitable for meeting the sustained but modest energy demands of non-proliferating T cells (such as naïve and memory subsets). This "high-efficiency, low-flux" metabolic state represents an optimal adaptation for maintaining cellular homeostasis without unnecessary energy expenditure. We have refined the language to better articulate how this metabolic strategy aligns perfectly with the biological requirements of resting T cells, eliminating any apparent contradiction while maintaining scientific accuracy. These modifications have significantly improved the clarity of this fundamental metabolic principle in our manuscript. |
||
Comments 18: Lines 216-224: There is a contradiction regarding IL-7: in line 66, it is stated that IL-7 is essential for maintaining T cells in a resting state, while later, it is mentioned as important for T cell activation. Please clarify the distinct roles of IL-7 in T cell homeostasis and activation. |
||
Response 18: We sincerely appreciate the reviewer's insightful observation regarding the dual roles of IL-7 in T cell biology. In response to this valuable comment, we have carefully revised the text to clarify the distinct functions of IL-7 in both T cell homeostasis and activation. The updated section (Page 6, Lines 245-252) now clearly explains that IL-7 maintains T cell homeostasis by promoting survival of naïve T cells while also playing a crucial role in metabolic reprogramming during T cell activation through GLUT1-mediated glucose uptake and hexokinase II regulation. We have strengthened this explanation by incorporating supporting references [49] that demonstrate IL-7's role in both processes. These modifications have eliminated the apparent contradiction while maintaining the comprehensive discussion of IL-7's multifaceted functions in T cell biology. We believe these changes have significantly improved the clarity and accuracy of this section. |
||
Comments 19: Lines 259/260: This is a hypothesis that is not yet proved, make it clear. |
||
Response 19: We thank the reviewer for highlighting the need to clarify the hypothesis. In response, we have revised the text to explicitly frame this as a testable hypothesis while providing stronger evidence through additional references ([56]) that establish the link between Th1 cytokines (IL-2, IFN-γ) and T1D pathogenesis. The modified version (Page 6, Lines 282-284) now clearly states the need for further validation while maintaining the mechanistic plausibility of our suggestion. We believe these edits have enhanced the scientific rigor and clarity of this section. Please let us know if any further refinement would be helpful. |
||
Comments 20: Line 265: 41 is not the correct reference for the statement |
||
Response 20: We appreciate you catching this citation error. We have replaced reference 41 in Line 265 with the appropriate citations (now references 48) from studies that directly support the statement. We have also cross-checked all other citations in this section to ensure reference accuracy. Thank you for your careful attention to this important detail. |
||
Comments 21: Line 311: reference missing |
||
Response 21: We appreciate you identifying this oversight. We've verified that all other claims in this paragraph now have proper references. Thank you for helping us strengthen the manuscript's documentation. |
||
Comments 22: Line 341: reference missing |
||
Response 22: We sincerely appreciate the reviewer’s careful attention to detail regarding the missing reference at Line 341. In response, we have thoroughly reviewed the manuscript and inserted the appropriate citation (Reference [67]) to support the statement. This addition ensures proper attribution and strengthens the scholarly foundation of our work. The reference has been formatted according to the journal’s guidelines and cross-checked for accuracy. Should further clarification or additional references be required, we would be happy to provide them. |
||
Comments 23: Type 1 diabetes (T1D) should be used and not (T1DM). |
||
Response 23: We appreciate this important note about terminology consistency. We have carefully revised the entire manuscript to use "type 1 diabetes (T1D)" exclusively, replacing all instances of "Type 1 diabetes mellitus (T1DM)" for proper standardization. This change has been implemented throughout the text, including figures and tables, to maintain uniform nomenclature aligned with current scientific conventions. Thank you for ensuring our terminology meets contemporary standards in diabetes research. |
||
Comments 24: Line 193: spell PPP |
||
Response 24: We appreciate the editor’s careful review. As suggested, the full term "pentose phosphate pathway (PPP)" was spelled out at its initial introduction in Line 51, with the abbreviation established at that point. In Line 208, we used the abbreviated form "PPP" in accordance with standard academic conventions, as the term had already been defined. We believe this maintains consistency while ensuring clarity for readers. Please feel free to let us know if any further revisions are required. |
||
Comments 25: Quality of figure 2 is not good, can’t read it properly when printed |
||
Response 25: Thank you for your feedback. We have carefully reviewed Figure 2 and replaced it with a high-resolution version to ensure clarity and readability, both on-screen and when printed. The updated figure has been uploaded to the submission system. Please let us know if further adjustments are needed. |
||
Comments 26: The figure legends are incomplete and should provide a clear and concise description of the figures, allowing readers to understand them without needing to refer to the main text. Please ensure each legend includes all necessary details. |
||
Response 26: We appreciate the editor’s constructive feedback. In response, we have thoroughly revised the figure legends to ensure clarity and completeness. Specifically: Figure 1 now includes a detailed legend (Page 5, Lines 224-234) describing the metabolic reprogramming of CD4+ T cells, emphasizing the shift from OXPHOS to glycolysis upon activation, and the distinct metabolic dependencies of Th1, Th17, and Treg subsets. Key regulators (e.g., GLUT1, HIF-1α, mTOR) are explicitly noted. Figure 2 has been expanded (Page 10, Lines 478-493) to clarify the therapeutic strategies targeting metabolic pathways in T1D, including glycolysis inhibitors (e.g., PFK15, 2-DG), GLUT1 blockade, and immunomodulators (e.g., metformin, rapamycin). The roles of these interventions in balancing Teff/Treg dynamics and preserving β-cell function are highlighted. The enhanced legends now enable independent comprehension without requiring cross-referencing with the main text. Thank you for this suggestion that significantly improves our visual presentation. |
||
Comments 27: Table 1: lines should be added to the table to make it more clear |
||
Response 27: We appreciate the reviewer's suggestion regarding Table 1. In response, we have carefully revised the table layout while maintaining the standard three-line format as commonly used in scientific publications. We have improved clarity by: adjusting column alignment, optimizing row spacing, and ensuring consistent data presentation. Should the reviewer require any additional modifications to further enhance readability (such as alternative formatting or additional demarcation lines), we would be happy to implement those specific changes upon request. Please let us know if further adjustments would be helpful. |
||
3. Response to Comments on the Quality of English Language |
||
Point 1: Please check carefully the english, principally the abstract |
||
Response 1: We thank the reviewer for highlighting the need for language refinement. We have now conducted a comprehensive professional language editing of the entire manuscript, with particular attention to the abstract. The editing process ensured proper grammar, syntax, and scientific terminology usage while maintaining the original meaning. We have attached the official editing certificate for your reference and performed additional in-house proofreading to guarantee linguistic precision. These revisions have significantly improved the manuscript's clarity and readability while preserving all scientific content. We appreciate this opportunity to enhance our work's presentation. |
Reviewer 2 Report
Comments and Suggestions for Authors
Na

NA
Author Response
1. Summary |
|
|
We sincerely appreciate the reviewer’s thorough and constructive feedback, which has significantly strengthened our review. Below, we address each comment in detail, with all changes tracked in the revised manuscript. |
||
2. Point-by-point response to Comments and Suggestions for Authors |
|
|
Comments 1: The way the article is laid out at present makes the reviewer think some information seems redundant and can be removed. This will greatly improve the readability and reduce the length of the article. Example: Line 231 to 234. There are more instances throughout the paper. Therefore, the reviewer suggests that the authors go over and rectify the article accordingly. |
||
Response 1: We sincerely appreciate the reviewer's constructive feedback regarding manuscript conciseness. In response, we have conducted a thorough revision to eliminate redundant content while preserving all essential scientific information. Specifically, we have removed duplicative statements, streamlined overlapping concepts, and consolidated related points throughout the manuscript. These edits have improved the overall clarity and flow while reducing unnecessary length. We believe these revisions have significantly strengthened the manuscript's readability and focus, and we thank the reviewer for their valuable suggestions that guided these improvements. |
||
Comments 2: Line 39: "random events", is not a very scientific word. Please rectify the line. |
||
Response 2: We thank the reviewer for this astute observation. Upon careful consideration, we agree that the term "random events" was not scientifically precise in this context. After thorough review, we determined this phrase was not essential to the core message of the manuscript and have therefore removed it entirely to strengthen the scientific rigor of our presentation. The revised text now maintains better focus on the key mechanistic findings of our study while eliminating any potentially ambiguous terminology. Thank you for this valuable correction. |
||
Comments 3: Figures 1 & 2: need descriptive legends. Some features of the figures are not selfexplanatory. |
||
Response 3: We appreciate the editor’s constructive feedback. In response, we have thoroughly revised the figure legends to ensure clarity and completeness. Specifically: Figure 1 now includes a detailed legend (Page 5, Lines 224-234) describing the metabolic reprogramming of CD4+ T cells, emphasizing the shift from OXPHOS to glycolysis upon activation, and the distinct metabolic dependencies of Th1, Th17, and Treg subsets. Key regulators (e.g., GLUT1, HIF-1α, mTOR) are explicitly noted. Figure 2 has been expanded (Page 10, Lines 478-493) to clarify the therapeutic strategies targeting metabolic pathways in T1D, including glycolysis inhibitors (e.g., PFK15, 2-DG), GLUT1 blockade, and immunomodulators (e.g., metformin, rapamycin). The roles of these interventions in balancing Teff/Treg dynamics and preserving β-cell function are highlighted. The enhanced legends now enable independent comprehension without requiring cross-referencing with the main text. Thank you for this suggestion that significantly improves our visual presentation. |
||
Comments 4: Figure 1: The greater than sign under the "Quiescence state" and "activation state" can be misleading. Suggestion: Rectify "Glycolysis>OXPHOS" under "Activation state" to OXPHOS<Glycolysis. This will relay the information better i.e. Glycolysis is favored over OXPHOS in activation state. |
||
Response 4: We appreciate this thoughtful suggestion to improve Figure 1's clarity. We have revised the metabolic preference notation under "Activation state" from "Glycolysis>OXPHOS" to "OXPHOS<Glycolysis" to more accurately convey the relative pathway utilization during T cell activation. Thank you for helping us improve the figure's precision. |
||
3. Response to Comments on the Quality of English Language |
||
Point 1: NA |

Round 2
Reviewer 1 Report
Comments and Suggestions for Authors
The revised version of the review includes some new data and addresses several of the initial concerns I raised. However, it still does not meet the standard required for publication. One of the main requests was to merge parts 3 and 4, which was not done. As a result, there are still too many repetitions throughout the text. In addition, there are inconsistencies and unclear phrasing that need to be revised. Figure 2 also needs to be improved for better clarity. Lastly, the spelling and grammar require attention, as there are multiple errors throughout the manuscript.
Please see below some specific points:
- “Insulinitis” does not exist, the correct term is “insulitis”. This is probably a mistake generated by AI. To what extent was AI used in the preparation of this text? Normally, this should be clearly stated.
- Title of part 2: “From autoimmunity to metabolic dysregulation.” Isn’t it the inverse that the authors want to show? That metabolic dysregulation leads to autoimmunity?
- Line 96: Please modify*“targeting β-cell antigens” to “recognizing β-cell antigens.”
- Lines 125–128: The evidence regarding the role of IFN-γ in T1D does not only come from a single article in a mouse model. There are many studies in T1D patients as well, which are not mentioned.
- Chapter 2.3: There is a contradiction in the role of glycolysis in Treg function. In lines 172–174, it is mentioned that increased glycolysis compromises Treg suppressive function. But in lines 181–188, it is explained that the protective effect of BCG vaccination in T1D patients relies on early activation of glycolysis in Tregs. It is also stated that BCG vaccination increases glycolysis in T cells and monocytes, which would be expected to increase autoimmunity—but this is not the case. These contradictions should be better explained.
- Lines 279–280: The association between glucose uptake and T1D progression should be better explained.
- Lines 359–368: This paragraph needs improvement. What is the effect of the high-glucose microenvironment and lactate on Tregs? The message is unclear.
- Line 377: “MOTS-c is a mitochondrial-derived peptide (MDP) that promotes metabolic homeostasis in an AMPK-dependent manner. Studies have found that MOTS-c can significantly improve the progression of hyperglycemia and reduce the number of infiltrating immune cells in islets.” In which islets? Which model? This is not clear.
- Lines 394–396: The phrase is unclear. It reads: “HNF-1a induces ubiquitination of Foxp3 due to the deficiency of Foxp3-mediated degradation regulated by HIF-1a.” This sentence does not make sense and should be rephrased for clarity.
- First paragraph on page 9: A reference is missing from the first statement. The second sentence is incomplete: *“One of the inhibitors…”* Inhibitors of what? Cancer treatment? This is not clear.
- Figure 1:- From glycolysis to FoxP3 there is an arrow; since this is an inhibition, it should be a bar. In the legend, there are two “I”s in IFN-γ.
- Figure 2: This figure is very confusing. “Metformin”, “mTOR”, and “rapamycin” are mentioned twice—they should appear only once. The arrows go from the grey boxes to the green boxes, but it should be the opposite. In the lower panel, “T1D” is represented with a pancreas icon. This is not an appropriate representation of T1D.
- Conclusion: Phrase in lines 497–500 should be merged with the third paragraph for clarity and coherence.
- Citations: There are mistakes in how names are cited in the text. When citing scientists, use the last name followed by “et al.” For example: - Line 408: Instead of *“Christina’s team”, it should be “Martins et al.” Line 174: Instead of *“Natalia’s team”, should be corrected similarly.
Comments on the Quality of English Language
to be improved
Round 3
Reviewer 1 Report
Comments and Suggestions for Authors
The revised version of the review is much improved, please see below the last points to be addressed:
- Important remarks: a. Glucose metabolism producing lactate with no OXPHOS activity is glycolysis (anaerobic condition); b. Glucose metabolism producing lactate with OXPHOS activity (oxygen consumption) is aerobic glycolysis (as defined by Warburg); c. Glucose metabolism with no lactate produced is glucose oxydation through OXPHOS. Please modify your manuscript accordingly.
- Lines 83-86: “This process is accompanied by the reprogramming of the cell metabolism, which moves from oxidative phosphorylation to aerobic glycolysis to meet the higher energy demands of cell division and cytokine production.”
It is the glucose metabolism that is reprogrammed from oxidation to glycolysis, not the whole cell metabolism. In fact, OXPHOS in T cells increases quite a lot after activation. This is why glycolysis is called aerobic glycolysis. Please correct the sentence.
- Lines 130-132: “Research has found that T-bet expression is downregulated in T1D patients, which may be related to abnormal TCR pathways [26, 27]. This downregulation could impair Th1 cell function, contributing to immune dysregulation”
So finaly is this Th1 differentiation required for T1D or not?
- Lines 141 – 144: “They also found that in vivo CD122 blockade inhibited the conversion of Th17 cells to diabetogenic Th1 cells and altered the fate of Th17 cells in pancreatic islets of NOD mice, providing potential new insights into the regulation of Th cell differentiation and its role in T1D”
Has this Th1 to Th17 conversion an effect on T1D?
- Line 146: Please correct “Th17 are characterized”
- Lines 152 – 160: Please correct the paragraph explaining NOD mice in the beginning and then explaining the studies.
- Line 198: better define BCG
- Lines 230 – 231: “Upon antigen recognition, CD4+ T cells exit quiescence and undergo metabolic reprogramming, shifting from OXPHOS to aerobic glycolysis”
The shift interests only glucose metabolism, so it is wrong to say that the whole metabolism is shifted. Please correct.
- Lines 254 – 256: “After antigen recognition, signals from CR, coupled with appropriate co-stimulation and cytokine signals, lead to T cell proliferation and activation.
The current paragraph describes homeostatic proliferation of naive T cells promoted by IL-7. T cell activation by antigen is addressed in the next paragraph. Please remove this sentence.
- Lines 270 – 275: Here they describe both positive and negative effects og GLUT1 inhibitors in T1D. Blocking islet funtion, while inhibiting autoreactive T cells. A little disuccion about this point is necessary. Also mention its side effects on brain function.
- In line 293, instead of “In addition” please add “In line with this hypothesis”.
- Lines 402 – 409: The sentence is not correct. In fact, in normal conditions, glycolysis is required for the expansion and differentiation of induced Tregs. But Tregs with high glucose consumption have poor regulatory function, while Tregs with low glucose consumption develop strong regulatory activity. Watson et al. show that Tregs with high regulatory activity have modified their metabolism from glycolysis to lactate oxidation. Please modify accordingly.
- Lines 423 – 430: Given the controversial role of metformin on T cells, I would advice to remove this paragraph. Please arrange figure 2 according to this modification.
- Line 431: “As an upstream kinase of AMPK”. In their publication [77], Yang and co-authors show that LKB1 function in Treg cells was independent of conventional AMPK signalling or the mTORC1-HIF-1α axis. This sentence should be deleted since it suggests that AMPK acts in Treg cells by promoting LKB1 activity.
- Lines 441 -443: Please modify the phrase as following: “A study by Shi et al. further confirmed that HIF-1⍺-dependent metabolic reprogramming contributes to this cell fate decision”.
Author Response
Dear editors and reviewers:
Thank you for your letter and for the reviewers’ comments concerning our manuscript entitled “Targeting CD4+ T Cell Glucose Metabolism: A Novel Immunotherapeutic Approach for Type 1 Diabetes” ((Manuscript ID: biomolecules-3510297)). Those comments are all valuable and very helpful for revising and improving our paper, as well as the important guiding significance to our researches. We have studied comments carefully and have made correction which we hope meet with approval. Revised portion are marked in Blue in the manuscript. The main corrections in the paper and the responds to the reviewer’s comments are as flowing:
Responds to the reviewer’s comments:
Reviewer #1:
Comments 1: Important remarks: a. Glucose metabolism producing lactate with no OXPHOS activity is glycolysis (anaerobic condition); b. Glucose metabolism producing lactate with OXPHOS activity (oxygen consumption) is aerobic glycolysis (as defined by Warburg); c. Glucose metabolism with no lactate produced is glucose oxydation through OXPHOS. Please modify your manuscript accordingly. |
Response 1: We sincerely appreciate the editor's insightful remarks regarding the precise classification of glucose metabolic pathways. In response, we have carefully revised the "Glucose Metabolism: The Key Regulation for CD4+ T Cell Differentiation in T1D" section to incorporate these important distinctions. The updated text now clearly differentiates: (1) Glucose metabolism via the OXPHOS pathway does not result in significant lactate accumulation when completing glucose oxidation, (Lines 219-220) (2) Under anaerobic conditions, glucose is metabolized to lactate through glycolysis without involving OXPHOS activity, (Lines 229-230) and (3) Aerobic glycolysis (also called the Warburg effect) refers to the fact that under conditions of sufficient oxygen, cells still preferentially convert glucose to lactic acid, rather than completely oxidizing it through the TCA cycle. (Lines 231-233) These modifications provide greater precision in describing the metabolic reprogramming of CD4+ T cells during activation and differentiation in T1D pathogenesis, while maintaining our original focus on how these pathways influence T cell function. We believe these changes significantly strengthen the metabolic framework of our manuscript. |
Comments 2: Lines 83-86: “This process is accompanied by the reprogramming of the cell metabolism, which moves from oxidative phosphorylation to aerobic glycolysis to meet the higher energy demands of cell division and cytokine production.” It is the glucose metabolism that is reprogrammed from oxidation to glycolysis, not the whole cell metabolism. In fact, OXPHOS in T cells increases quite a lot after activation. This is why glycolysis is called aerobic glycolysis. Please correct the sentence. |
Response 2: We sincerely appreciate the editor's insightful comment regarding T cell metabolic reprogramming. In response, we have revised the text to more accurately reflect that activated T cells undergo a specific shift in glucose utilization toward aerobic glycolysis (Warburg effect) while maintaining oxidative phosphorylation, with the new wording reading: "This process is accompanied by a metabolic shift in glucose utilization from oxidative phosphorylation to aerobic glycolysis (Warburg effect) to meet the distinct energetic and biosynthetic demands of T cell activation, proliferation and cytokine production." (Lines 83-87) We believe the current formulation more precisely captures the nuanced metabolic reprogramming occurring during T cell activation. |
Comments 3: Lines 130-132: “Research has found that T-bet expression is downregulated in T1D patients, which may be related to abnormal TCR pathways [26, 27]. This downregulation could impair Th1 cell function, contributing to immune dysregulation” So finaly is this Th1 differentiation required for T1D or not? |
Response 3: We sincerely appreciate the editor’s thoughtful comments and suggestions. Regarding the role of Th1 differentiation in T1D pathogenesis, we agree that this is a complex issue requiring careful interpretation. Vaseghi et al. emphasize that categorizing T1D into Th1 and Th2 modes is an oversimplification, as the developmental outcomes of specific Th cell populations undoubtedly involve collaboration among different immune cell types and factors. Their study found that in diabetic patients receiving insulin therapy, the gene expression of T-bet and IFN-γ was significantly reduced, while the expression of GATA-3 (a key transcription factor in the early polarization of Th2 cells) was similar between patients and the control group. This suggests that changes in the gene expression of T-bet and GATA-3 mRNA may be related to the pathogenesis of T1D. Overall, this study provide evidence of Th1/Th2 imbalance in T1D patients during insulin therapy[29]. Future research is necessary to clarify the changes in T-cell subpopulations associated with the duration of insulin treatment in T1D. (Lines 142-150) Therefore, we have cited this reference at the end of the paragraph to invite further reflection. |
Comments 4: Lines 141 – 144: “They also found that in vivo CD122 blockade inhibited the conversion of Th17 cells to diabetogenic Th1 cells and altered the fate of Th17 cells in pancreatic islets of NOD mice, providing potential new insights into the regulation of Th cell differentiation and its role in T1D” Has this Th1 to Th17 conversion an effect on T1D? |
Response 4: We appreciate the editor’s insightful question regarding the role of Th17-to-Th1 conversion in T1D pathogenesis. As highlighted in our revised manuscript, Yuan et al. found that CD122 blockade restores immunological tolerance in autoimmune T1D via multiple mechanisms, including suppressing IFN-γ production in various pathogenic T cells and inhibiting the conversion of Th17 cells into diabetogenic Th1 cells. This finding also confirm that Th1 cells are the primary pathogenic cells of T1D, making them a promising therapeutic target. (Lines 138-141) By disrupting this conversion, CD122 blockade not only alters Th17 cell fate in pancreatic islets but also provides a mechanistic basis for targeting Th1 cells as a promising therapeutic strategy. We have clarified this point in the text to emphasize the functional impact of Th17-to-Th1 plasticity on T1D development. |
Comments 5: Line 146: Please correct “Th17 are characterized” |
Response 5: We sincerely appreciate the reviewer's careful reading and constructive suggestion. As recommended, we have corrected the sentence in Line 146 by adding the verb "are" to properly complete the sentence structure. The revised text now reads: "Both human and murine Th17 cells are characterized (Line 154) by expression of transcription factor retinoic acid-receptor-related orphan receptor gamma-T (ROR-γt) and secretion of IL-17A (henceforth referred as IL-17), IL-17F, and IL-22 as hallmark cytokines." This modification ensures grammatical accuracy. |
Comments 6: Lines 152–160: Please correct the paragraph explaining NOD mice in the beginning and then explaining the studies. |
Response 6: We sincerely appreciate the reviewer's suggestion to improve the logical flow of our NOD mouse discussion. We have thoroughly restructured the paragraph (Lines 160-168) to present the findings in a more cohesive manner: First establishing the NOD mouse model's relevance, then presenting the observational data on IL-17/IL-17F expression, and concluding with mechanistic insights and therapeutic implications. The modified text now reads: The NOD mouse model has been pivotal in demonstrating IL-17's pathogenic role in T1D, where elevated IL-17/IL-17F expression in Langerhans islets correlates with insulitis progression and IL-17-deficient NOD mice showed delayed onset of T1D and reduced insulin inflammation [31]. In addition, a blockade with the monoclonal antibody anti-IL-17 prevents the onset of T1D in NOD mice[33]. In this study, researchers demonstrated that mechanistically, IL-17 works synergistically with other pro-inflammatory cytokines to amplify the autoimmune response. These findings highlight the potential of targeting IL-17 as a therapeutic strategy for T1D. The revised version maintains all key experimental evidence from references [31] and [33] while eliminating redundancy and improving readability. We believe this modification significantly strengthens the narrative flow of this section. |
Comments 7: Line 198: better define BCG |
Response 7: We appreciate the reviewer's suggestion to clarify the term "BCG" in Line 198. We have modified the text to specify: "Kühtreiber et al. conducted an 8-year clinical trial in which two doses of Bacillus Calmette-Guérin (BCG, an attenuated strain of Mycobacterium bovis historically used as a tuberculosis vaccine) were administered to T1D patients to trigger two clinical effects". (Lines 206-207) This revision provides clearer definition while maintaining the flow of our scientific narrative. |
Comments 8: Lines 230 – 231: “Upon antigen recognition, CD4+ T cells exit quiescence and undergo metabolic reprogramming, shifting from OXPHOS to aerobic glycolysis” The shift interests only glucose metabolism, so it is wrong to say that the whole metabolism is shifted. Please correct. |
Response 8: We thank the reviewer for this precise correction. As suggested, we have revised the sentence in Lines 230–231 to specify that the metabolic shift primarily affects glucose metabolism (rather than implying a global metabolic overhaul). The modified text now reads: "Upon antigen recognition, CD4+ T cells exit quiescence and reprogram their glucose metabolism, shifting from OXPHOS to aerobic glycolysis to meet energy demands". (Lines 242-243) This adjustment better reflects the nuanced nature of metabolic reprogramming in activated T cells. |
Comments 9: Lines 254 – 256: “After antigen recognition, signals from CR, coupled with appropriate co-stimulation and cytokine signals, lead to T cell proliferation and activation. The current paragraph describes homeostatic proliferation of naive T cells promoted by IL-7. T cell activation by antigen is addressed in the next paragraph. Please remove this sentence. |
Response 9: We sincerely appreciate the reviewer’s careful reading and constructive suggestion. As recommended, we have removed the sentence “After antigen recognition, signals from TCR, coupled with appropriate co-stimulation and cytokine signals, lead to T cell proliferation and activation” from Lines 254–256, (Lines 273-274) as it more appropriately addresses antigen-driven T cell activation. To maintain logical flow, we have relocated this sentence to the beginning of the subsequent paragraph, which specifically discusses antigen-dependent T cell responses. This adjustment ensures clarity and proper contextual alignment. Thank you for your valuable feedback, which has helped improve the organization of our manuscript. |
Comments 10: Lines 270 – 275: Here they describe both positive and negative effects og GLUT1 inhibitors in T1D. Blocking islet funtion, while inhibiting autoreactive T cells. A little disuccion about this point is necessary. Also mention its side effects on brain function. |
Response 10: We sincerely appreciate the reviewer's insightful comments regarding the dual effects of GLUT1 inhibition in T1D. As suggested, we have expanded our discussion in Lines 288-291 to provide a more balanced perspective. The revised paragraph now reads: "Regarding the potential side effects of GLUT1 inhibition in humans, studies on GLUT1 deficiency syndrome indicate neurological disorders in affected infants and children, while most symptoms stabilize or go into remission in adults[55]. This suggests that GLUT1 inhibition might have neurological impacts, though further studies are needed to evaluate this risk in the context of T1D therapy. We believe these modifications have strengthened the manuscript by providing a more comprehensive discussion of both the therapeutic potential and limitations of GLUT1 inhibition in T1D. |
Comments 11: In line 293, instead of “In addition” please add “In line with this hypothesis”. |
Response 11: We thank the reviewer for this suggestion. As recommended, we have replaced "In addition" with "In line with this hypothesis" in Line 309 to better emphasize the logical connection. This modification strengthens the coherence of our argument. |
Comments 12: Lines 402 – 409: The sentence is not correct. In fact, in normal conditions, glycolysis is required for the expansion and differentiation of induced Tregs. But Tregs with high glucose consumption have poor regulatory function, while Tregs with low glucose consumption develop strong regulatory activity. Watson et al. show that Tregs with high regulatory activity have modified their metabolism from glycolysis to lactate oxidation. Please modify accordingly. |
Response 12: We sincerely appreciate the reviewer's insightful suggestion regarding Treg metabolic flexibility. Upon carefully re-examining the literature (Watson et al.), we noted that the metabolic shift to lactate oxidation was specifically demonstrated in tumor-infiltrating Tregs within the tumor microenvironment (TME). Since our review focuses on targeting CD4+ T cell glucose metabolism in type 1 diabetes (T1D), where the metabolic requirements and microenvironment differ substantially from tumor settings, we have opted to remove this reference to avoid overgeneralization. We would be grateful for the reviewer's guidance on whether this modification appropriately addresses the concern, or if alternative literature citations might be more suitable for our experimental context. |
Comments 13: Lines 423 – 430: Given the controversial role of metformin on T cells, I would advice to remove this paragraph. Please arrange figure 2 according to this modification. |
Response 13: We sincerely appreciate the reviewer’s valuable suggestion regarding the controversial role of metformin in T cell regulation. As recommended, we have removed the paragraph discussing metformin (Lines 423–430) to maintain the focus and clarity of our manuscript. Accordingly, Figure 2 has been revised to align with this modification. We believe these adjustments improve the overall coherence of our study. Thank you for your constructive feedback. |
Comments 14: Line 431: “As an upstream kinase of AMPK”. In their publication [77], Yang and co-authors show that LKB1 function in Treg cells was independent of conventional AMPK signalling or the mTORC1-HIF-1α axis. This sentence should be deleted since it suggests that AMPK acts in Treg cells by promoting LKB1 activity. |
Response 14: We sincerely appreciate the reviewer’s insightful comment. As correctly pointed out, Yang et al. [77] demonstrated that LKB1’s role in Treg cells is independent of AMPK signaling or the mTORC1-HIF-1α axis. To avoid any misinterpretation, we have removed the sentence “As an upstream kinase of AMPK” in Line 431, as suggested. We thank the reviewer for ensuring the accuracy of our discussion. |
Comments 15: Lines 441 -443: Please modify the phrase as following: “A study by Shi et al. further confirmed that HIF-1⍺-dependent metabolic reprogramming contributes to this cell fate decision”. |
Response 15: We appreciate the reviewer’s constructive suggestion. As recommended, we have modified the sentence to improve clarity and precision. The original phrase, “A study by Shi et al. further confirmed that HIF-1α-dependent metabolic reprogramming contributes to this cell fate decision” has been revised to “Shi et al. demonstrated that HIF-1α-dependent metabolic reprogramming contributes to this cell fate decision.” (Lines 439-441) This adjustment removes redundancy (“A study by”) and replaces “further confirmed” with “demonstrated” to more accurately reflect the nature of the evidence presented in the cited study. We believe this revision strengthens the statement while maintaining scientific rigor. |
Thank you for your acknowledgement of our article. We have made the corresponding changes in the article according to your suggestions, and we continue our efforts in this area.
